# MULTI-AGENT REINFORCEMENT LEARNING WITH SHARED RESOURCES FOR INVENTORY MANAGEMENT

## ABSTRACT

In this paper, we consider the inventory management (IM) problem where we need to make replenishment decisions for a large number of stock keeping units (SKUs) to balance their supply and demand. In our setting, the constraint on the shared resources (such as the inventory capacity) couples the otherwise independent control for each SKU. We formulate the problem with this structure as Shared-Resource Stochastic Game (SRSG) and propose an efficient algorithm called Context-aware Decentralized PPO (CD-PPO). Through extensive experiments, we demonstrate that CD-PPO can accelerate the learning procedure and achieve better performance compared with standard MARL algorithms.

## 1 INTRODUCTION

The inventory management (IM) problem has long been one of the most important application scenarios in the supply-chain industry (Nahmias & Smith, 1993). Its main purpose is to maintain a balance between the supply and demand of stock keeping units (SKUs) in a supply chain by optimizing the replenishment decisions for each SKU. An efficient inventory management strategy cannot only increase the profit and reduce the operational cost but also give rise to better services to help maintain customer satisfaction (Eckert, 2007). Nevertheless, this task is quite challenging in practice due to the fact that the replenishment decisions for different SKUs compete for shared resources (e.g., the inventory capacity or the procurement budget) as well as cooperate with each other to achieve a high total profit. This becomes more challenging when the number of SKUs involved in the supply-chain becomes larger. Such co-existence of cooperation and competition renders IM a complicated and challenging problem.

Traditional methods usually reduce the IM problem to dynamic programming (DP). However, these approaches often rely on unrealistic assumptions such as *iid* customer demands and deterministic leading time (Kaplan, 1970; Ehrhardt, 1984). Moreover, when the state space grows rapidly along with the scaling-up of some key factors such as the leading time and the number of SKUs, the problem becomes intractable by DP due to the curse of dimension (Gijsbrechts et al., 2019). Due to these limitations, many approaches based on approximate dynamic programming are proposed to solve IM problems in different settings (Halman et al., 2009; Fang et al., 2013; Chen & Yang, 2019). While these approaches perform well in certain scenarios, they rely heavily on problem-specific expertise or assumptions, e.g., the zero or one period leading time assumption in (Halman et al., 2009), and thus can hardly generalize to other settings. In contrast, reinforcement learning (RL) based methods, with short inference time, can be generalized into various scenarios in a data-driven manner. However, it is hard to train a global policy that makes decisions for all SKUs due to the large global state and action space (Jiang & Agarwal, 2018). To further address the training efficiency issue, it is natural to adopt the multi-agent reinforcement learning (MARL) paradigm, where each SKU is controlled by an individual agent whose state and action spaces are localized and only contain information relevant to itself.

There are currently two popular paradigms to train MARL in the literature: independent learning (Tan, 1993) and joint action learning (Lowe et al., 2017). Despite of their success in many scenarios, these two MARL paradigms also exhibit certain weaknesses that restrain their effectiveness in solving the IM problem. For independent learning, the policy training of one agent treats all other agents as a part of the stochastic environment. It may largely increase the hardness of training convergence due to the non-stationarity of the environment. For joint action learning, a centralized critic is usu-

ally trained to predict the value based on the global state (of all SKUs) and the joint action, which can easily become intractable with a growing number of SKUs. Furthermore, it is time-consuming to sample data from a joint simulator for a large number of SKUs due to the high computational cost to calculate the internal variables that model the complex agent interactions.

To address these challenges, we leverage the structure in the IM problem to design a more effective MARL paradigm. Particularly, each agent in system interacts with the others only through the competence of shared resources such as the inventory capacity. We introduce an auxiliary variable called *context* to represent the shared resources (e.g., the available inventory level for all SKUs). From the MARL perspective, the dynamics of the context actually reflect the collective behaviors of all the agents. Conditioned on the context dynamics, we assume the transition dynamics and the reward function of the agents are independent. In this way, leveraging the context as an additional input for the policy or the value network of each agent enables us to not only prevent the non-stationarity in independent learning but also feed global information to the critic without leading to an intractable global critic.

Based on the context, we propose Shared-Resource Stochastic Game (SRSG) to model the IM problem. Since the context and the policies of each agent depend on each other, it is hard to solve for them simultaneously. Accordingly, we make two assumptions to circumvent this issue: 1) rearranging the sampling process by first sampling the contexts and then sampling local the state/action/reward for each agent; 2) using context dynamics sampled by previous policies. With these assumptions, we can design an efficient algorithm called Context-aware Decentralized PPO (CD-PPO) that consists of two iterative learning procedures: 1) obtaining the context samples from a joint simulator, and 2) updating the policy for each agent by the data sampled from its corresponding local simulator conditioned on the collective context dynamics. By decoupling each agent from the others with a separate local simulation, our method can greatly reduce the model complexity and accelerate the learning procedure. At last, we conduct extensive experiments and the results validate the effectiveness of our method. Besides, not limited to the IM problem considered in this paper, our method may be applied to other applications with shared resources such as portfolio management (Ye et al., 2020) where different stocks share the same capital pool and smart grid scheduling (Remani et al., 2019) where different nodes share a total budget.

Our contributions are summarized as follows:

- We propose Shared-Resource Stochastic Game to capture the problem structure in the IM problem where agents interact with each other through competing for shared resources.

- We propose a novel algorithm called Context-aware Decentralized PPO that leverages the shared-resource structure to solve the IM problem efficiently.

- We conduct extensive experiments to demonstrate that our method can achieve the performance on par with state-of-the-art MARL algorithms while being more computationally and sample efficient.

## 2 BACKGROUND

### 2.1 STOCHASTIC GAMES

We build our work on the formulation of the stochastic game (SG) (Shapley, 1953) (also known as Markov game). A stochastic game is defined as a tuple $(\mathcal{N}, \mathcal{S}, \mathcal{A}, \mathcal{T}, R, \gamma)$ where $\mathcal{N} = [n]$ denotes the set of $n > 1$ agents, $\mathcal{S}$ is the state space, $\mathcal{A} := \mathcal{A}^1 \times \cdots \times \mathcal{A}^n$ is the action space composed of the action spaces of individual agents, $\mathcal{T} : \mathcal{S} \times \mathcal{A} \to \Delta(\mathcal{S})$ is the transition dynamics, $R = \sum_{i=1}^{n} R^i$ is the total reward which is the summation of individual rewards $R^i : \mathcal{S} \times \mathcal{A}^i \times \mathcal{S} \to \mathbb{R}$ and $\gamma \in [0, 1)$ is the discount factor. For the $i$-th agent, we denote its policy as $\pi^i : \mathcal{S} \to \Delta(\mathcal{A}^i)$ and the joint policy of the other agents as $\pi^{-i} = \prod_{j \in [n] \setminus i} \pi^j$. Each agent optimizes its policy conditioned on the policies of the others, i.e.,

$$\max_{\pi^i} \eta^i(\pi^i, \pi^{-i}) = \mathbb{E}\left[\sum_{t=0}^{\infty} \gamma^t r_t^i \Big| \pi^i, \pi^{-i}\right], \tag{1}$$

where $r_t^i$ is the individual reward obtained by following the joint policy composed of $\pi^i$ and $\pi^{-i}$. We will later propose Shared-Resource Stochastic Game (SRSG) which is a special case of stochastic game.

## 2.2 INVENTORY MANAGEMENT WITH SHARED RESOURCE

In this paper, we consider a simplified setting with one store and multiple SKUs to better focus on the shared-resource structure of the problem. We further assume that there is an upstream warehouse that can fulfill all the requirements from the store. Our objective is to train a high-quality replenishing policy for each SKU in the store, particularly when there are a large number of SKUs. It is worthwhile to mention the following two points: First, this setting is more challenging than managing the SKUs in warehouses due to the direct impact of the changing customer demands. Second, due to the flexibility of RL algorithms, our method can also be applied to more complex settings, e.g., with multi-echelon structure or fluctuate supply.

Similar to previous work, we follow the multi-agent RL (MARL) framework with decentralized agents, each of which manages the inventory of one SKU in the store. We assume that the store has $n$ SKUs in sell, all of which share a common inventory capacity that can store up to $I_{\max}$ units at the same time.

Replenishing decisions of each SKU are made on discretized time steps (which are days in this paper). For the $t$-th time step and the $i$-th SKU, we denote the in-stock quantity by $\dot{I}_t^i \in \mathbb{Z}$. This quantity is constrained by the limit of the inventory capacity (i.e., the shared resources). Accordingly, the following constraint holds:

$$\forall t \geq 0, \sum_{i=1}^n \dot{I}_t^i \leq I_{\max}. \tag{2}$$

The dynamics of the system is as follows: For the $t$-th time step and the $i$-th SKU, the corresponding agent can place a replenishment order to request $O_t^i \in \mathbb{Z}$ units of products from its upstream warehouse. It takes several time steps (referred to as the leading time $L_t^i$) before these products are delivered to the store. We denote the total units in transit at the $t$-th time step by $T_t^i \in \mathbb{Z}$ and the total units arrived at the $t$-th time step by $A_t^i \in \mathbb{Z}$. Meanwhile, the customer demand is $D_t^i$ and leads to an actual sale of $S_t^i \in \mathbb{Z}$ units. Due to the possibility of out-of-stock, $S_t^i$ may be smaller than $D_t^i$. We consider the setting where the leading time $L_t^i$ and the demand $D_t^i$ are stochastic.

Since all the SKUs share a common inventory space, the storage may overflow when the previously ordered products arrive. In this paper, we assume that the excess SKUs will be discarded proportionally according to a coefficient $\rho$ which is the overflowing ratio if we accept all the arriving products. To calculate $\rho$, we introduce an extra variable $\hat{I}_t^i$ which is the afterstate of $\dot{I}_t^i$. Intuitively, it denotes how many units of the $i$-th SKU in stock at the end of the $t$-th time step if we omit the capacity constraint. Note that, although other ways to deal with overflow (e.g., prioritizing some SKUs) are possible, our algorithm will also apply to these settings and currently we only consider this naive way. Undoubtedly, overflow will lead to extra operational cost which should be avoided as much as possible in the replenishing strategy.

For clearness, we summarize the dynamics these variables as follows:

$$S_t^i = \min\left(D_t^i, \dot{I}_t^i\right) \qquad A_t^i = \sum_{\tau:\tau+L_\tau^i=t} O_\tau^i$$

$$T_{t+1}^i = T_t^i - A_{t+1}^i + O_{t+1}^i \qquad \hat{I}_t^i = \dot{I}_t^i - S_t^i + A_{t+1}^i$$

$$\rho_t = \max\left[\sum_{i=1}^n \hat{I}_t^i - I_{\max}, 0\right] \Big/ \left[\sum_{i=1}^n \hat{I}_t^i\right] \tag{3}$$

$$\dot{I}_{t+1}^i = \dot{I}_t^i - S_t^i + \lfloor(1-\rho_t)A_{t+1}^i\rfloor \tag{4}$$

The immediate profit of the $i$-th SKU is calculated according to the following equation:

$$F_t^i = p_i S_t^i - q_i O_t^i - o\mathbb{I}\left[O_t^i > 0\right] - h\dot{I}_t^i \tag{5}$$

where $p_i$ and $q_i$ are the unit sale price and the unit procurement price for the $i$-th SKU respectively, $o$ and $h$ are the order cost and the unit holding cost respectively, and $\mathbb{I}[\cdot]$ is an indicator function which equals to one when the condition is true and zero otherwise. The order cost reflects the fixed transportation cost or the order processing cost, and yields whenever the order quantity is non-zero. See the further description for the problem setting in Appendix A.

## 3  METHODOLOGY

In this section, we will first introduce the Shared-Resource Stochastic Game (SRSG) formulation which decouples the control of individual agents in the multi-agent problem with shared resources by introducing *context*. Next, we introduce the dynamics of the *context* and how to build an efficient local simulator that receives the context trajectories as the input for the IM problem considered in our paper. To accelerate the sampling process in estimating and optimizing the objective function, we further introduce two approximations which leads to a practical algorithm called Context-aware Decentralized PPO (CD-PPO).

### 3.1  SHARED-RESOURCE STOCHASTIC GAME

In this section, we formulate the multi-agent problem with shared resources as Shared-Resource Stochastic Game (SRSG) where each agent is influenced by the others only through a shared resource limit. We define the shared-resource stochastic game as a tuple $\left(\mathcal{N}, \{\mathcal{S}^i\}_{i \in \mathcal{N}}, \mathcal{C}, \mathcal{A}, \mathcal{T}, R, \gamma\right)$ where the definitions of $\mathcal{N}$, $\mathcal{A}$ and $\gamma$ follow the definitions in the stochastic game introduced in Section 2.1. We use $\mathcal{S}^i$ to denote the local state space of the $i$-th agent, and $\mathcal{C}$ to denote the *context* space observed by all agents. For example, the *context* can be used to represent the occupied capacity of shared resources (e.g., the inventory space). Denote $\mathcal{S} := \mathcal{S}^1 \times \cdots \times \mathcal{S}^n$ and $\mathcal{A} := \mathcal{A}^1 \times \cdots \times \mathcal{A}^n$. The transition dynamics $\mathcal{T} : \mathcal{S} \times \mathcal{C} \times \mathcal{A} \to \Delta(\mathcal{S} \times \mathcal{C})$ can be decomposed as the context transition dynamics $c_{t+1} \sim P_c(\cdot \mid c_t, s_t, a_t)$ and the local state transition dynamics $s_{t+1}^i \sim P_s^i(\cdot \mid s_t^i, a_t^i, c_{t+1})$ where $c_t \in \mathcal{C}$, $s_t^i \in \mathcal{S}^i$, $s_t \in \mathcal{S}$, $a_t^i \in \mathcal{A}^i$, and $a_t \in \mathcal{A}$. The reward function $r_t = \sum_{i=1}^n r_t^i$ is additive and $r_t^i \sim R^i(s_t^i, c_t, a_t^i)$. The objective for the $i$-th agent is to learn a policy $\pi^i : \mathcal{S}^i \times \mathcal{C} \to \Delta(\mathcal{A}^i)$ to maximize the objective function that shares the same form as that defined in equation 1.

Given the above definition, an IM problem can be formulated as SRSG by letting $r_t^i = F_t^i$, $a_t^i = O_t^i$, and $c_t = \sum_{i=1}^n \dot{I}_t^i$ for all $i \in [n]$ and time step $t$. Moreover, the state of the $i$-th SKU is a concatenation of information such as the in-transit quantity $T_t^i$, the sale price $p_i$, its historical actions and the customer demands (see details in Appendix C).

### 3.2  CONTEXT DYNAMICS AND LOCAL SIMULATOR.

We use $\dot{c}_t^i$ to represent the amount of resources occupied by the $i$-th agent at the $t$-th time step and $\dot{c}_t$ to represent the total amount of occupied resources. We further use $\dot{c}_t^{-i}$ to denote the amount of resource occupied by all agents except $i$. Assuming that the resources are additive, we have the following equations:

$$\dot{c}_t = \sum_{i=1}^n \dot{c}_t^i, \qquad \dot{c}_t = \dot{c}_t^i + \dot{c}_t^{-i}. \tag{6}$$

Similarly, we denote the afterstate of $\dot{c}_t^i$ as $\hat{c}_t^i$, i.e., the amount of occupied resources at the end of the $t$-th time step. From the perspective of the $i$-th agent, the context $\dot{c}_t^{-i}$ can be regarded as a part of the environment. Hence, $P(\hat{c}_t^i \mid s_t^i, a_t^i, \dot{c}_t^i)$ represents how the replenishment decision influences $\hat{c}_t^i$. Similarly, we have $\hat{c}_t = \hat{c}_t^i + \hat{c}_t^{-i}$. Then, by applying a resource overflow resolution procedure defined in equation 3 and equation 4, we can obtain $\dot{c}_{t+1}^i$ (the resources in the next step).

We use the notation $c_t$ to represent $(\hat{c}_{t-1}, \dot{c}_t)$. We refer $(s^i, c^i)$ as the state for the $i$-th agent, and $c^{-i}$ as the context. For each agent and given the context trajectory $\{c_\tau^{-i}\}_{\tau \geq 0}$, the sampling process

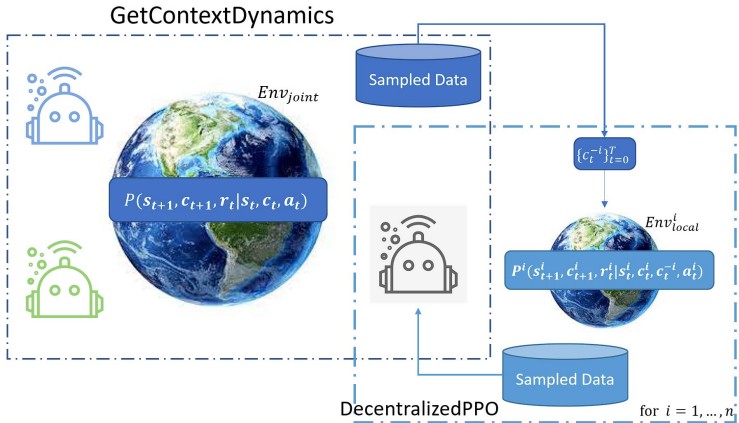

Figure 1: Our algorithm consists of two iterative learning procedures: 1) Collect the context trajectory $\{c_t^{-i}\}_{t=0}^T$ from the joint simulator with previous policies $\pi_{\text{old}}^i$, $\pi_{\text{old}}^{-i}$ for each $i \in [n]$. 2) Train each policy $\pi^i$ with samples collected from the local simulator conditioned on context trajectory $\{c_t^{-i}\}_{t=0}^T$.

(denoted as $\mathcal{T}^i$) is as follows:

$$
\begin{aligned}
c_{t+1}^i &\sim P_c^i(\cdot \mid s_t^i, a_t^i, c_t^i, c_t^{-i}), \\
s_{t+1}^i &\sim P_s^i(\cdot \mid s_t^i, a_t^i, c_{t+1}^i, c_{t+1}^{-i}), \\
r_t^i &\sim R^i(s_t^i, a_t^i, c_{t+1}^i, c_{t+1}^{-i}).
\end{aligned}
\tag{7}
$$

Accordingly, we can build a local simulator for each agent according to the above equations. Note that the local simulator is computationally more efficient than the joint simulator that simulates the process of all SKUs simultaneously. To simplify the presentation, we only consider one kind of shared resource (i.e., the inventory space) in the above formulation. However, it can be extended to support multiple resources as long as these resources are additive.

For the tractability and the computational efficiency of the sampling process for evaluating the objective function defined in equation 1, we make two approximations to the sampling process.

First, we assume that each agent has infinitesimal influence on the context $c^{-i}$. This is reasonable for the shared-resource setting where the resource consumption of one agent does not significantly influence that of the others. Second, we assume the policy changes slowly during the learning process. Therefore, instead of repeatedly using the current policies to collect the context trajectories, we allow to use the context trajectories collected by the policies in the previous iterations. Accordingly, we can run the joint simulator with the current policies infrequently to collect trajectories $\{c_\tau^{-i}\}_{\tau \geq 0}$ for different agents indexed by $i$. Then, we can evaluate the local rewards with the local simulator under different possible local policies for each agent. In this way, we can greatly reduce the frequency to simulate using the joint simulator which is time-consuming.

## 3.3 ALGORITHM

In the subsection, we will present details of our proposed algorithm called Context-aware Decentralized PPO (CD-PPO). We call it *context-aware* because we feed the context $c_t^{-i}$ as the input of the policy and the value function of each agent. Such a context-aware approach can avoid the non-stationary problem in independent learning methods (which do not consider the policy change of the other agents during training) since the context reflects the collective behaviors of the other agents which can in turn impact the dynamics of each individual agent. In the meanwhile, our method can mitigate the centralized critic (i.e., the critic that receives $(s_t, a_t, c_t)$ as the input) in many centralized training and decentralized execution (CTDE) methods which becomes intractable when the number of agents increases and the joint state and action space explodes.

We show our algorithm in Algorithm 1 and Figure 1, which consists of two iterative learning procedures: 1) obtaining the context trajectories by running all agents in the joint environment; 2) updating the policy of each agent using the data sampled from its local simulator conditioned on the context

---

**Algorithm 1** Context-aware Decentralized PPO

---
Given the joint simulator $\text{Env}_{\text{joint}}$
Given the local simulators $\{\text{Env}_{\text{local}}^i\}_{i=1}^n$ for each agent
Initialize the policy $\pi^i$ and the value function $V^i$ for each agent $i = 1, \dots, n$
**for** $m = 1, 2, \cdots, M$ epochs **do**
    // Collect context trajectories from joint simulation
    $\{c_t^{-1}\}_{t=0}^T, \dots, \{c_t^{-n}\}_{t=0}^T \leftarrow$ **GetContextDynamics**$(\text{Env}_{\text{joint}}, \{\pi^i\}_{i=1}^n)$
    **for** $k = 1, 2 \dots, K$ **do**
        **for all** $i = 1, 2, \cdots, n$ agents **do**
            // Set the context trajectory
            $\text{Env}_{\text{local}}^i.\text{set\_context\_trajectory}(\{c_t^{-i}\}_{t=0}^T)$
            // Train policy by simulating in the corresponding local environment
            $\pi^i, V^i \leftarrow$ **DecentralizedPPO**$(\text{Env}_{\text{local}}^i, \pi^i, V^i)$
        **end for**
    **end for**
**end for**

---

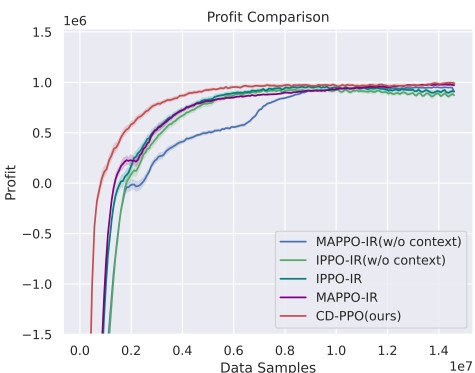

Figure 2: Training curves of different algorithms in the setting with 5 SKUs. "IR" indicates that we train the agents using only individual rewards and "w/o context" indicates that we do not feed the global information related to the shared resources to the agent. The x-axis takes the samples from all local simulators into account for CD-PPO[2].

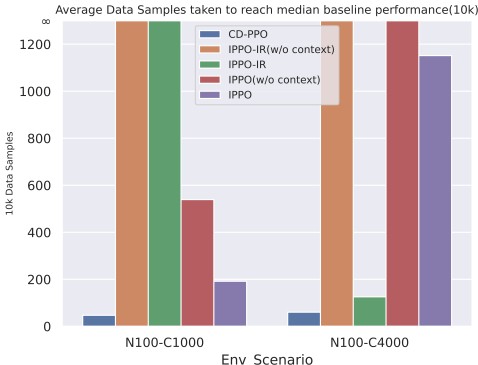

Figure 3: The average number of samples needed by different algorithms to reach the median performance across all baselines in the setting with 100 SKUs. The lower the value, the higher the sample-efficiency for the algorithm. "N" and "C" denote the number of SKUs and the maximum capacity of shared resource respectively.

trajectories. Moreover, our algorithm follows a decentralized training paradigm with infrequent communication through the context trajectories. Due to space limit, we refer readers to Appendix B for a detailed description of the algorithm (including the description of the two subroutines).

It is worth noting that a naive approach is to optimize policies based on the samples collected from the joint simulator. Nonetheless, we find that it takes more computational costs to simulate for one step in the joint simulator than simulating for one step in each of the $n$ local simulators due to the complex agent interaction. Such phenomena become more significant when there are more types of shared resources or more complex constraints to consider (e.g., when we consider a smarter discarding strategy to deal the overflow instead of discarding excess units for each SKU according to $\rho_t$). In our experiments, the simulator we are using is developed for the general purpose. It contains extra details such as replenishing orders fulfillment scheduling and transportation management, and thus requires large computational costs to simulate the IM problem with more SKUs. Our method, with the advantage of leveraging local simulators to accelerate simulation, is more computationally efficient.

## 4 EXPERIMENTS

We evaluate the performance of CD-PPO in several inventory management problem instances with different scales (i.e., with 5, 50, 100, and 1000 SKUs) and different resources limits (by changing the limit of the inventory space). We demonstrate that, our algorithm can achieve better results and is significantly more sample-efficient than the previous SOTA baselines.

**Setups.** Our experiment is conducted on a simulator that can simulate the replenishment procedure for multiple SKUs in one store. Instead of sampling demands from some hypothetical distributions, we replay the demand series from two authentic datasets: a dataset from Kaggle (Makridakis et al., 2020) that contains the sales history of five years (2011-2016) with 155 randomly selected SKUs and a dataset from a retail partner that contains the sales history of two years (2018-2019) with 1000 SKUs. The former dataset is used to support the experiment settings with 5, 50, and 100 SKUs and the latter dataset is used for the setting with 1000 SKUs. We use the sales of the last one year (the last 100 days) as the testing set for the former dataset (for the latter dataset) and the remaining data as the training set. For other necessary information (such as the prices, the costs, and the leading time) to instantiate our simulator but not included in the dataset, we first select a reasonable range for each of the parameters and then sample from the range. We use the total profit in dollars as the evaluation metric. We run all the algorithms with four different random seeds and present the average performance with standard deviations.

**Baselines.** The baseline algorithms used in our experiment is based on the EPyMARL (Papoudakis et al., 2021) framework where commonly used MARL algorithms are implemented. In our preliminary experiments, we find that two model-free on-policy MARL algorithms IPPO (de Witt et al., 2020) and MAPPO (Yu et al., 2021) outperform the other algorithms including COMA (Foerster et al., 2018), QMIX (Rashid et al., 2018), etc. MAPPO learns a centralized critic and therefore becomes intractable due to the large state-action space when the number of SKUs is greater or equal to 50. Due to the bad performance of the other algorithms (i.e., mostly resulting in negative profits), we only show the experiment results of different variants of IPPO (and MAPPO only for the setting with 5 SKUs). We also compare with two operation research (OR) methods: the base-stock policy (Kaplan, 1970) and the $(s, S)$-policy (Ehrhardt, 1984) which are two well-known OR strategies and are widely adopted in practice (see e.g., Kapuscinski & Tayur, 1999; Hubbs et al., 2020). Previous studies make great efforts to determine the parameters in these strategies. We will present their performance with the optimal parameters for each SKU obtained based on the future information, which can be regarded as the skyline. (See details in Appendix D.)

**Implementation.** We share the parameters of the policy network and the value network across different agents/SKUs. The policy/value network consists of a two-layer MLP with 64 neurons on each hidden layer. In our experiment, the policy network maps the local state to a categorical distribution over a discrete action space $\{0, \frac{1}{3}, \frac{2}{3}, 1, \frac{4}{3}, \frac{5}{3}, 2, \frac{5}{2}, 3, 4, 5, 6, 7, 9, 12\}$ where the real replenishment quantity is obtained by multiplying these numbers with the average daily sales of the SKU over the past two weeks. By default, the RL algorithms use the individual reward rather than the team reward (e.g., the summation of the individual rewards from all the agents) as the objective of each agent. We also feed global information related to the shared resources to the agents. In CD-PPO, to encourage the policy to adapt for stochastic context dynamics in practice, we adopt the following techniques in the training phase to augment the collected context trajectories: 1) adding noise to randomly chosen items from context trajectories, or 2) replacing randomly chosen items with predicted values generated by a context generating model. (See details on the implementation in Appendix C.)

**Results.** For the scenario with 5 SKUs, we show the training curves in Figure 2. We can observe that while CD-PPO converges to the same performance level as the other algorithms in this simple setting, it is more sample-efficient due to its efficient learning with local simulators.

To evaluate our algorithm on larger scenarios, we run CD-PPO and IPPO on the settings with 50, 100, and 1000 SKUs. To evaluate sample efficiency, we also record the number of samples when different algorithms reach the median performance level across all baselines. We summarize the results in Table 1 and Figure 3. The experiment results indicate that CD-PPO outperforms other

---

[2]Specifically, for one interaction in the global environment with $N$ agents, we consider it as $N$ samples. For one interaction in the local simulator (based on context trajectories) with one agent, we consider it as 1 data sample. We empirically observe that the running time is roughly proportional to this statistics.

Table 1: Comparison of the profits from different algorithms on the settings with different number of SKUs (indicated by "N") and different limits of the inventory space (indicated by "C"). "IR" indicates training with individual reward and "NC" indicates feeding the global information related to the shared resources to the agent. Base-stock and $(s, S)$-policy are the skylines.

|  | N50C500 | N50C2k | N100C1k | N100C4k | N1000C30k | N1000C200k |
|---|---|---|---|---|---|---|
| IPPO-IR-NC | 235±61 | 689±49 | -2107±315 | -2223±2536 | 88±87 | 571±102 |
| IPPO-IR | 250±58 | 546±460 | -1126±410 | 148±1017 | 120±71 | 366±213 |
| IPPO-NC | 164±143 | -1373±870 | -1768±1064 | -6501±6234 | -1966±1871 | -2495±3874 |
| IPPO | **367±90** | -1103±1116 | -670±1396 | -6019±9056 | -186±222 | -7±99 |
| CD-PPO | 311±76 | **695±174** | **660±150** | **1298±125** | **233±197** | **741±157** |
| Base-stock | 688 | 919 | 1189 | 1887 | 73 | 322 |
| (s,S)-policy | 704 | 941 | 1227 | 1981 | 621 | 1210 |

RL baselines on 5 out of the 6 settings and even beats the performance of the base-stock policy skyline. (Recall that we search the best parameters for each SKU using the future information in these skylines.) The superior performance of CD-PPO may due to the avoidance of non-stationarity by introducing the context. In contrast, traditional MARL methods with the CTDE (centralized training and decentralized execution) paradigm cannot scale up to the environment with even 50-100 SKUs due to the fact that the inputs of the centralized critic is too large to be loaded in to the memory. IPPO, on the other hand, can run successfully but underperforms CD-PPO. More importantly, we observe that CD-PPO is more sample efficient than IPPO. Notice that, since there is no need for the agents to be synchronized before the next time running the joint simulation, the training for individual agents can be parallelized to further reduce the wall time. The full results and more ablation studies on how the capacity of shared resource affects the performance of CD-PPO and the influence of the context augmentation technique can be found in Appendix E.

## 5 RELATED WORK

In this section, we will introduce the related work on the inventory management problem and the training paradigms in MARL.

**Inventory management.** The inventory management problem has a long history and the classical methods for single-echelon systems include the economic order quantity (EOQ) model (Harris, 1913; Wilson, 1934), the $(R, Q)$-policy (Chen, 2000), the base-stock policy (Clark & Scarf, 1960; Kaplan, 1970), the $(s, S)$-policy (Ehrhardt, 1984), and other continuous/periodic review methods (cf. Axsäter, 2015). The base-stock policy is a special case of the $(s, S)$-policy and we provide a detailed introduction for these two methods in Appendix D. The $(s, S)$-policy and the $(R, Q)$-policy are proved to be optimal under different assumptions (see Chen, 2000). However, these assumptions do not hold in many practical scenarios and researchers start to solve the inventory management problem using the approximate form (Halman et al., 2009; Fang et al., 2013; Chen & Yang, 2019) of dynamic programming (DP) instead of the exact form (Huh et al., 2009; Goldberg et al., 2016). Using the DP solution based on posterior information as the label to train an end-to-end model achieves superior performance in real industrial scenarios (Qi et al., 2020).

To better exploit the patterns behind big data, reinforcement learning becomes a popular choice for solving the inventory management problem in a data-driven manner (see e.g., Giannoccaro & Pontrandolfo, 2002; Jiang & Sheng, 2009; Kara & Dogan, 2018; Barat et al., 2019; Gijsbrechts et al., 2019; Oroojlooyjadid et al., 2017; 2020). However, as their main focus is to deal with challenges such as volatile customer demands and bullwhip effects in IM, they are restricted to simplified scenarios with only a single SKU. While these approaches are able to outperform classical methods in these scenarios, they overlook the system constraints and the coordination between the SKUs constrained by shared resources. Exceptions are two recent papers (Barat et al., 2019; Sultana et al., 2020) where more realistic scenarios containing multiple SKUs are considered. In Barat et al. (2019), the main contribution is to propose a framework to support efficient deployment of RL algorithms in real systems. As an example, the authors introduce a centralized algorithm for solving the IM problem. In contrast, a decentralized algorithm is proposed in Sultana et al. (2020) to solve the IM problem with multiple SKUs in a multi-echelon setting. These two papers use the off-the-shelf RL algorithm (i.e., A2C Wu et al., 2018) to solve for the policy.

**Training paradigms in MARL.** From the perspective of the training paradigms in MARL, the algorithms can be divided into three categories: centralized training methods, centralized training and decentralized execution (CTDE) methods, and independent learning methods. The centralized training methods adopt single agent RL algorithms directly to the multi-agent setting (see e.g., Gupta et al., 2017). However, these methods can only learn a centralized policy which is not suitable for scenarios that call for policies to be executed in a decentralized way. Accordingly, the CTDE methods are more preferable and they can be further divided into 1) value-based CTDE methods (aka value decomposition methods) that centrally learn a value function that can be reasonably decoupled into decentralized ones such as VDN (Sunehag et al., 2017), QMIX (Rashid et al., 2018), Weighted QMIX (Rashid et al., 2020), and QTRAN (Son et al., 2019); and 2) policy-based CTDE methods that also relies on a centralized critic such as COMA (Foerster et al., 2018), MADDPG (Lowe et al., 2017), and MAPPO (Yu et al., 2021). However, for the inventory management problem considered in our paper, there are a large number of agents in the system in which the centralized critic becomes intractable due to the large joint state space. Therefore, only independent learning methods are applicable to our setting. A representative independent learning MARL algorithm is IPPO (de Witt et al., 2020) where each agent learns a critic and a policy (with parameters shared with others) based on the local observations to predict values and make individual decisions respectively. However, these methods easily suffer from the non-stationarity problem since each agent has to learn in an environment that changes across the training process without the information on how the policies of the others change. In this paper, we propose Context-aware Decentralized PPO (CD-PPO) which follows the independent learning paradigm (i.e., without the need to resort to the joint state/action space) but avoids the non-stationarity problem by feeding a context variable to the agent.

Weakly-coupled MDP is first proposed by (Meuleau et al., 1998) to model the scenario where the constraints of the shared global or local resources couple the otherwise independent sub-processes. However, their solution is closely related to the air campaign planning problem and can hardly generalize to other applications. Our setting is similar to weakly-coupled MDP with shared local resources where the instant (instead of the cumulative) inventory level cannot exceed a certain limit, but they still differ since the shared local resources directly result in a feasible action set in their setting which is easier to solve. The subsequent work on weakly-coupled MDP solves the problem with classical optimization methods such as approximate linear program or Lagrangian relaxation (see e.g., Adelman & Mersereau, 2008; Ye et al., 2017; Brown & Zhang, 2022), whereas our algorithm is a practical deep reinforcement learning based method for the multi-agent setting with limits on shared resources. Mean-field MARL (Yang et al., 2018; Carmona et al., 2019; Chen et al., 2021) is derived from the seminal work on mean-field games (Lasry & Lions, 2007; Guéant et al., 2011) to model the scenario where there exists a large number of interchangeable/indistinguishable agents and the effect of each agent on the system is infinitesimal. This setting is different from ours where the optimal policy for each agent is different and each agent learns based on a local context variable $c_t^{-i}$ instead of a global mean-field variable.

## 6 CONCLUSION

In this paper, we consider the inventory management problem for a single store with a large number of SKUs. In our setting, the control for different SKUs interact indirectly with each other through a constraint on the limit of the shared resources. We first modeled the problem as Shared-Resources Stochastic Game (SRSG) to leverage the shared-resource structure of the problem. Starting from SRSG, we proposed Context-aware Decentralized PPO (CD-PPO) where each agent can be trained independently in a local simulator conditioned on the context trajectories that screen the interactions from other agents. Our experiment results indicated that CD-PPO outperforms state-of-the-art MARL algorithms while being more computationally sample efficient. It is worth mentioning that our method is not only limited to the inventory management problem but also has the potential to apply to other real-world multi-agent applications with a similar shared-resource structure.

In real-world applications, we usually need to deal with thousands of agents, which poses a challenge for existing MARL algorithms. To address the challenge, we need to develop efficient algorithms that enables effective (e.g., being stable during the training) and efficient (e.g., with a distributed training paradigm) training. In this paper, we took our first step towards developing efficient and practical MARL algorithms for real-world applications with the shared-resource structure, and we will continue to address the challenges arisen in real-world applications in our future work.

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

## A  DETAILS FOR THE INVENTORY MANAGEMENT PROBLEM

In this appendix, we summarize the notations introduced in Section 2.2 in Table 2. We also give an example with two SKUs in Figure 4 to further illustrate the dynamics of the inventory management problem.

## B  DETAILS FOR THE ALGORITHM

In this section, we will introduce the details of the designs in CD-PPO including the details for the subroutines in Algorithm 1. Additionally, to augment the context trajectories collected by the joint simulator, we design a context augmentation technique which leads to a variant of CD-PPO with context augmentation (cf. Algorithm 4). We will provide experiment results on this variant in Appendix E.

### B.1  GET CONTEXT DYNAMICS

Our algorithm follows MAPPO (Yu et al., 2021) and maintains two separate networks for $\pi_\theta$ and $V_\phi$.

In the first stage of our algorithm, we run the policies in the origin joint environment to get context trajectories, i.e., $\{c_t^{-i}\}_{t=0}^T$. This process is similar to other popular MARL methods. Note that we can also save the transitions of each agent for learning the policy and the value network. We provide the the pseudocode of this joint sampling process in Algorithm 2 where we record the context trajectories for the training in the next state.

Additionally, we design a context augmentation technique in this stage to augment the collected context trajectories. After collecting the context trajectories, we use an LSTM model $f_\mathbf{c}$ to train a surrogate predictor as an extra augmentation for the collected context trajectories in next stage. In details, we split the collected context trajectories into several sub-sequences of length 8 and the training objective is to minimize the mean-squared error of the $L+1$ day's capacity predicted given the dynamics of previous $L$ days.

$$\min(f_\mathbf{c}(c_{t-L}, \ldots, c_t; \omega) - c_{t+1})^2 \tag{8}$$

This results in CD-PPO with context augmentation which is presented in Algorithm 4.

### B.2  DECENTRALIZED PPO

With the collected context dynamics of the shared resource, it is easy to run the second stage: sampling on the local simulators for each agent and then training the policy and critic with data. It is worth noting that the main difference between our training paradigm and traditional MARL methods under CTDE structure is that we directly sampling local observations in the extra simulators in

Table 2: Notations for the inventory management problem.

| Notation | Explanation |
|---|---|
| $\dot{I}_t^i$ | Units in stock of SKU $i$ at the end of the $t$-th time step |
| $\hat{I}_t^i$ | Units in stock of SKU $i$ at the $t$-th time step before discarding the excess |
| $L_t^i$ | Lead time for the order placed at the $t$-th time step |
| $O_t^i$ | Order quantity of the $i$-th SKU at the $t$-th time step |
| $A_t^i$ | Arrived quantity of the $i$-th SKU at the $t$-th time step |
| $D_t^i$ | Demand of the $i$-th SKU at the $t$-th time step |
| $S_t^i$ | Sale quantity of the $i$-th SKU at the $t$-th time step |
| $T_t^i$ | Units in transit of the $i$-th SKU at the $t$-th time step |
| $F_t^i$ | Profit generated on the $i$-th SKU at the $t$-th time step |
| $p_i$ | Unit sales price of $i$-th SKU |
| $q_i$ | Unit procurement cost of $i$-th SKU |
| $o$ | Unit order cost |
| $h$ | Unit overnight holding cost |

which only one agent existed rather than the joint simulator in which all agents interacting with each other. In other words, in the new local simulator, there is only one SKU in the entire store, and the trend of available capacity is completely simulated according to the given context dynamics.

In practice, we parallelly initialize new instances of the original inventory environment with the new configure settings which only contains a specific SKU $i$ and a fixed trajectory of context. As for the fixed context trajectory, we use the subtraction results $\{c^{-i}\}_{t=0}^{T}$ with some augmentations: 1) add some noise in some items with a predefined probability;2) replace some items with predicted values comes from the trained context model also by the predefined probability. Then we run the policy to interact with the local simulators to conduct episodes under the embedded context dynamics and put them into a shared replay buffer since all transitions are homogeneous and we shared the parameters over all policies. And decentralized training will be started by utilizing the shared replay buffer of transitions collected from local simulators.

We consider a variant of the advantage function based on decentralized learning, where each agent learns a agent-specific state based critic $V_\phi(s_t^i)$ parameterized by $\phi$ using *Generalized Advantage Estimation* (GAE, (Schulman et al., 2016)) with discount factor $\gamma = 0.99$ and $\lambda = 0.95$. We also add an entropy regularization term to the final policy loss (Mnih et al., 2016). For each agent $i$, we have its advantage estimation as follows:

$$A_t^i = \sum_{l=0}^{h} (\gamma\lambda)^l \delta_{t+l}^i \tag{9}$$

where $\delta_{ti}^i = r_t\left(s_t^i, a_t^i\right) + \gamma V_\phi\left(z_{t+1}^i\right) - V_\phi\left(s_t^i\right)$ is the TD error at time step $t$ and $h$ is marked as steps num. And we use individual reward provided from local simulator $r_t^i(s_t^i, a_t^i)$. So that the final policy loss for each agent $i$ becomes:

$$
\begin{aligned}
\mathcal{L}^i(\theta) = \mathbb{E}_{s_t^i, a_t^i \sim \mathcal{T}_{\text{local}}(c_t^{-i})} \Bigg[ &\min\Bigg( \frac{\pi_\theta\left(a_t^i \mid s_t^i\right)}{\pi_{\theta_{old}}\left(a_t^i \mid s_t^i\right)} A_t^i, \\
&\text{clip}\left( \frac{\pi_\theta\left(a_t^i \mid s_t^i\right)}{\pi_{\theta_{old}}\left(a_t^i \mid s_t^i\right)}, 1-\epsilon, 1+\epsilon \right) A_t^i \Bigg) \Bigg]
\end{aligned}
\tag{10}
$$

As for training value function, in addition to clipping the policy updates, our method also use value clipping to restrict the update of critic function for each agent $i$ to be smaller than $\epsilon$ as proposed by GAE using:

$$
\begin{aligned}
\mathcal{L}^i(\phi) = \mathbb{E}_{s_t^i \sim \mathcal{T}_{\text{local}}(c_t^{-i})} \Bigg[ \min\Bigg\{ &\left( V_\phi\left(s_t^i\right) - \hat{V}_t^i \right)^2, \\
&\left( V_{\phi_{old}}\left(s_t^i\right) + \text{clip}\left( V_\phi\left(s_t^i\right) - V_{\phi_{old}}\left(s_t^i\right), -\epsilon, +\epsilon \right) - \hat{V}_t^i \right)^2 \Bigg\} \Bigg]
\end{aligned}
\tag{11}
$$

where $\phi_{old}$ are old parameters before the update and $\hat{V}_t^i = A_t^i + V_\phi\left(s_t^i\right)$. The update equation restricts the update of the value function to within the trust region, and therefore helps us to avoid

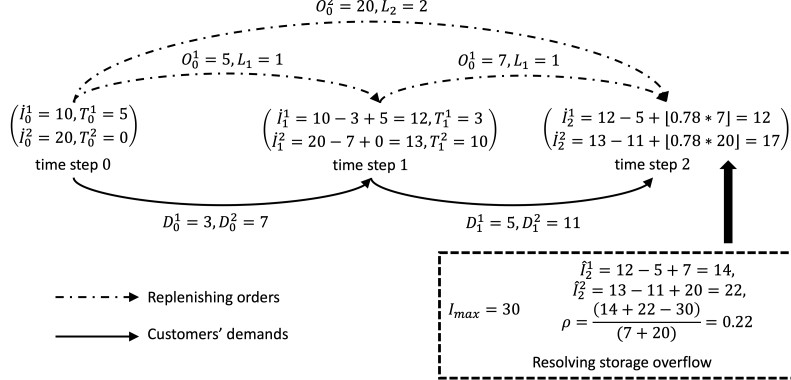

Figure 4: A diagram to illustrate an inventory dynamics in two time steps.

overfitting to the most recent batch of data. For each agent, the overall learning loss becomes:

$$\mathcal{L}(\theta, \phi) = \sum_{i=1}^{N} \mathcal{L}^i(\theta) + \lambda_{\text{critic}} \mathcal{L}^i(\phi) + \lambda_{\text{entropy}} \mathcal{H}\left(\pi^i\right) \tag{12}$$

It is obvious that all networks are trained in the decentralized way since their inputs are all local variables which stem from the light-cost local simulators. As mentioned before, at this learning stage, there are no interactions between any two agents. Although it seems like the way of independent learning, we need to point that we use the global context simulated from the joint environment, which is essentially different from independent learning methods since they will not consider this style global information which is simulated from joint simulator but be fixed in the local simulators. Our decentralized training have several advantages: firstly, the local simulator is running efficient because of its simple only-one-agent transition function; secondly, this paradigm avoid the issue for non-stationary occurred in the traditional MARL methods since there are no interaction amongst agents so that it is no need to consider influences of other agents; thirdly, we can use more diverse context trajectories to make agents face various situations of the available levels of the store, which leads to improve the generalization of the networks to be trained; fourthly, it is easy for this training paradigm to be extended to large-scale distributed training by running parallel simulation whose communication cost is also acceptable for modern distributed training frameworks.

If the critic and actor networks are RNNs, then the loss functions additionally sum over time, and the networks are trained via Backpropagation Through Time (BPTT). Pseudocode for local sampling with recurrent version of policy networks is shown in Algorithm 3.

---

**Algorithm 2** GetContextDynamic

---

**INPUT** policies $\{\pi_{\theta_i}^i\}_{i=1}^n$ and the joint simulator $\text{Env}_{joint}$
(Optional) Initialize $\omega$, the parameters for context model $f_{\mathbf{c}}$
set data buffer $D = \{\}$
**for** $bs = 1$ **to** $batch\_size$ **do**
   $\tau = []$ empty list
   initialize $h_{0,\pi}^1, \cdots h_{0,\pi}^n$ actor RNN states
   **for** $t = 1$ **to** $T$ **do**
      **for all** agents $i$ **do**
         $p_t^i, h_{t,\pi}^i = \pi^i(s_t^i, c_t, h_{t-1,\pi}^i; \theta_i)$
         $a_t^i \sim p_t^i$
      **end for**
      Execute actions $\boldsymbol{a_t}$, observe $\boldsymbol{r_t}, \boldsymbol{s_{t+1}}$
   **end for**
   // Split trajectory $\tau$ into chunks of length L
   **for** l = 0, 1, .., T//L **do**
      $D = D \cup (c[l : l + T])$
   **end for**
**end for**
// (Optional) Train the context model for augmentation
**if** Need to train context model **then**
   **for** mini-batch $k = 1, \ldots, K_1$ **do**
      $b \leftarrow$ random mini-batch from D with all agent data
      Update capacity-context dynamics model with Eq. 8
   **end for**
   Adam update $\omega$ with data $b$
**end if**
**OUTPUT** context dynamics $\{c_t^{-i}\}_{t=1}^T$ for $i = 1, \ldots, n$; (Optional) $f_\omega$

---

---

**Algorithm 3** DecentralizedPPO

---

// Generate data for agent $i$ with corresponding context dynamic model
**INPUT** local simulator $\text{Env}_{local}^i$, policy $\pi_{\theta_i}^i$ and value function $V_{\phi_i}^i$
Set data buffer $D = \{\}$
Initialize $h_{0,\pi}^1, \cdots, h_{0,\pi}^n$ actor RNN states
Initialize $h_{0,V}^1, \ldots h_{0,V}^n$ critic RNN states
$\tau = []$ empty list
**for** $t = 1$ **to** $T$ **do**
$\quad p_t^i, h_{t,\pi}^i = \pi^i(s_t^i, h_{t-1,\pi}^i, c_t^i; \theta_i)$
$\quad a_t^i \sim p_t^i$
$\quad v_t^i, h_{t,V}^i = V^i(s_t^i, c_t^i, h_{t-1,V}^i; \phi_i)$
$\quad$ Execute actions $a_t^i$ in $\text{Env}_{local}^i$, and then observe $r_t^i, c_{t+1}^i, s_{t+1}^i$
$\quad \tau + = [s_t^i, c_t^i, a_t^i, h_{t,\pi}^i, h_{t,V}^i, s_{t+1}^i, c_{t+1}^i]$
**end for**
Compute advantage estimate $\hat{A}$ via GAE on $\tau$ (Eq. 9)
Compute reward-to-go $\hat{R}$ on $\tau$
// Split trajectory $\tau$ into chunks of length L in $D$
**for** l = 0, 1, .., T//L **do**
$\quad D = D \cup (\tau[l : l + T, \hat{A}[l : l + L], \hat{R}[l : l + L])$
**end for**
**for** mini-batch $k = 1, \ldots, K_2$ **do**
$\quad b \leftarrow$ random mini-batch from D with all agent data
$\quad$ **for** each data chunk $c$ in the mini-batch $b$ **do**
$\quad\quad$ update RNN hidden states for $\pi^i$ and $V^i$ from first hidden state in data chunk
$\quad$ **end for**
**end for**
Calculate the overall loss according to Eq. equation 10-equation 12
Adam update $\theta_i$ on $L^i(\theta_i)$ and $gH$ with data $b$
Adam update $\phi_i$ on $L^i(\phi_i)$ with data $b$
**OUTPUT** policy $\pi_{\theta_i}^i$ and value function $V_{\phi_i}^i$

---

---

**Algorithm 4** Context-aware Decentralized PPO with Context Augmentation

---

Given the joint simulator $\text{Env}_{\text{joint}}$ and local simulators $\{\text{Env}_{\text{local}}^i\}_{i=1}^n$
Initialize policies $\pi^i$ and value functions $V^i$ for $i = 1, \ldots, n$
Initialize context model $f_\omega$ and the augmentation probability $p_{aug}$
**for** $M$ epochs **do**
$\quad$ // Collect context dynamics via running joint simulation
$\quad \{c_t^{-1}\}_{t=0}^T, \ldots, \{c_t^{-n}\}_{t=0}^T, f_\omega \leftarrow$ **GetContextDynamics**$(\text{Env}_{\text{joint}}, \{\pi^i\}_{i=1}^n, f_\omega)$ (Algorithm 2)
$\quad$ **for** $k = 1, 2 \ldots, K$ **do**
$\quad\quad$ **for all** agents $i$ **do**
$\quad\quad\quad$ // Set capacity trajectory by augmented context dynamics
$\quad\quad\quad \text{Env}_{\text{local}}^i.\text{set\_c\_trajectory}(aug(\{c_t^{-i}\}_{t=0}^T, f_\omega, p_{aug}))$
$\quad\quad\quad$ // Train policy by running simulation in the corresponding local environment
$\quad\quad\quad \pi^i, V^i \leftarrow$ **DecentralizedPPO**$(\text{Env}_{\text{local}}^i, \pi^i, V^i)$ (Algorithm 3)
$\quad\quad$ **end for**
$\quad$ **end for**
$\quad$ Evaluate policies $\{\pi^i\}_{i=1}^n$ on joint simulator $\text{Env}_{\text{joint}}$
**end for**

---

## C   DETAILS FOR THE EXPERIMENTS

### C.1   THE CODEBASE

As part of this work we extended the well-known EPyMARL codebase((Papoudakis et al., 2021)) to integrated our simulator and algorithm, which already include several common-used algorithms and support more environments as well as allow for more flexible tuning of the implementation details. It is convenience for us to compare our algorithm with other baselines. All code for our new codebase is publicly available open-source on Anonymous GitHub under the following link: `https://anonymous.4open.science/r/replenishment-marl-baselines-75F4`

### C.2   HYPERPARAMETERS DETAILS

We present the hyperparameters used in our algorithm for environment with 5 SKUs in Table 3.

Table 3: Hyparameters used in CD-PPO

| Hyperparameters | Value |
|---|---|
| runner | ParallelRunner |
| batch size run | 10 |
| decoupled training | True |
| use individual envs | True |
| max individual envs | 5 |
| decoupled iterations | 1 |
| train with joint data | True |
| context perturbation prob | 1.0 |
| hidden dimension | 64 |
| learning rate | 0.00025 |
| reward standardisation | True |
| network type | FC |
| entropy coefficient | 0.01 |
| target update | 200 (hard) |
| n-step | 5 |

### C.3   DETAILS OF STATES AND REWARDS

Table 4 shows the features of the state for our MARL agent that corresponds to the $i$-th SKU on the $t$-th time step.

It is worthy to note that we use the profit generated on the $i$-th SKU at the $t$-th time step $F_t^i$ divided by $1 \times 10^6$ as the individual reward of $i$-th agent at the $t$-th time step, for team reward methods, we simply sum up all the individual rewards, which corresponds to the daily profit of the whole store at the $t$-th time step, divided by $1 \times 10^6$.

## D   BASE STOCK POLICY AND (S, S)-POLICY

In this paper, we choose two well-known classical methods, the base-stock policy and the $(s, S)$-policy, from the OR community as our non-RL baselines. In the $(s, S)$-policy, the two parameters $s$ and $S$ control the replenishment strategy where $s$ is the reorder point and $S$ is the maximum level. On each time step (or review period), when the inventory position (including the units in stock and the units in transit) declines to or below the reorder point $s$, we order up to the maximum level $S$. The base-stock policy is a special case of the $(s, S)$-policy with $s = S$, i.e., on each time step we place an order to fill the inventory position up to $S$. Previous studies proposes different algorithms to determine these two parameters (see e.g., Axsäter, 2015). In our experiments, we search for the posteriori optimal parameters for each of the SKUs in control. Therefore, the performance of these two policies presented in Table 1 can be regarded as the best performance that can be achieved by such classical methods.

Table 4: Features of the state

| | Features |
|---|---|
| **State** | |
| Storage information | Storage capacity C |
| | |
| Inventory information | Quantity of products in stock $I_t^i$ |
| | Quantity of products in transit $T_t^i$ |
| | |
| History information | Replenishment history in the latest 21 days $\left[O_{t-21}^i, \cdots, O_{t-1}^i\right]$ |
| | Sales history in the latest 21 days $\left[S_{t-21}^i, \cdots, S_{t-1}^i\right]$ |
| | Standard deviation of historical sales $\mathrm{std}\left(S_{t-21}^i, \cdots, S_{t-1}^i\right)$ |
| | |
| Product information | Unit sales price $p_i$ |
| | Unit procurement cost $q_i$ |
| **Context** | |
| Global storage utilization | Current total storage level of the store $\sum_{j=1}^{l-1} I_t^j$ |
| Global unloading level | Current total unloading level of the store $\sum_{j=1}^{l-1} O_{t-L_j+1}^j$ |
| Global excess level | Current total excess level of the store $\rho \times \sum_{j=1}^{l-1} O_{t-L_j+1}^j$ |

# E ADDITIONAL EXPERIMENT RESULTS

## E.1 THE COMPLETE RESULTS

Here we present the complete experiment results in Table 5 and display the training curves of algorithms in 50-SKUs scenarios in Figure 5 and Figure 6. We also present the results from OR methods designed to deal with IM problem, namely the base-stock policy, in Table 5.

Please note that, actually, VLT (i.e., leading time) for each SKU in N50 and N100 is stochastic during simulation. On the one hand, the specified VLTs can be modelled as exponential distributions in our simulator. On the other hand, the real lead time for each procurement can be longer than specified due to lack of upstream vehicles. (Distribution capacity is also considered in the simulator, though not the focus of this paper). So that the results running on N50 and N100 are all considering stochastic VLTs.

It is worthy noting that throughout all the environments we have experimented on, CD-PPO enjoys higher sample efficiency as shown in Table 6. Though it seems that in environment where the storage capacity is extremely tense, like N5-C50, CD-PPO performs not as well as IPPO, we reason that these tense situations favor IPPO (with team reward), which may quickly learn how to adjust each agents' behavior to improve team reward. While CD-PPO, owing to its decentralized training paradigm, struggles to learn a highly cooperative policy to cope with the strong resource limits while meeting stochastic customer demands. Yet we still claim that CD-PPO does outperform its IR (w/o context) baselines while producing comparable performance when compared to IR (with context) algorithms. On the other hand, IPPO(with team reward) struggles to learn a good policy in 50-agents environment for the large action space compared to the single team reward signal, and it fails when faced with N50-C2000 whose state space is even larger. To top it all off, CD-PPO does have its strengths in terms of sample efficiency. Though we will continue to address the above challenge in our future work.

## E.2 ABLATION STUDIES FOR CONTEXT AUGMENTATION

In this section, we aim to seek a better way to augment the context dynamics in order to boost performance of our algorithm. We ponder the following two questions:

**Q1: Which is a better way to augment the original data, adding noise or using a Deep prediction model?**

To answer this question, we set out experiments on environment N5-C100 with our algorithm CD-PPO, using either context trajectories generated by a deep LSTM prediction model, or simply adding a random normal perturbation to the original dynamics trajectories, both on 3 different seeds. As

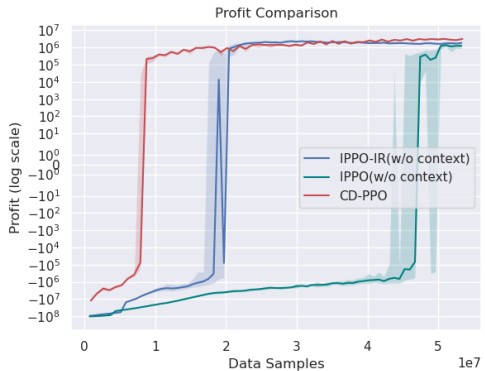
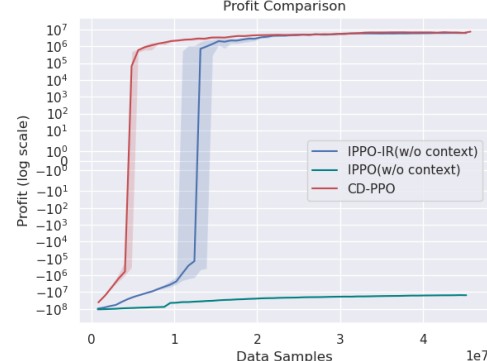

Figure 5: Training curves on N50-C500.   Figure 6: Training curves on N50-C2000

shown in Figure 7, those runs with deep prediction model generated dynamics enjoy less std and better final performance. This could result from that the diversity of deep model generated trajectories surpasses that of random perturbation.

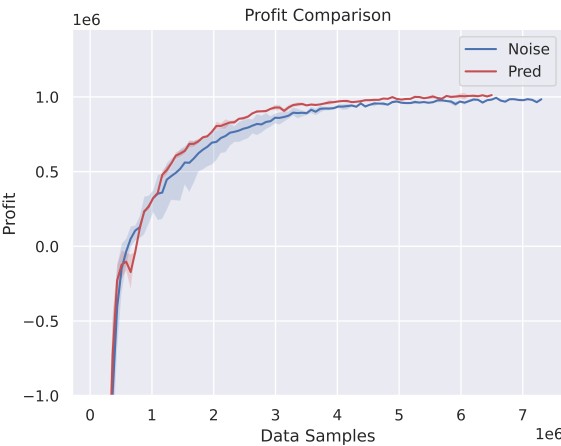

Figure 7: Training curves of CD-PPO with different augmentation methods

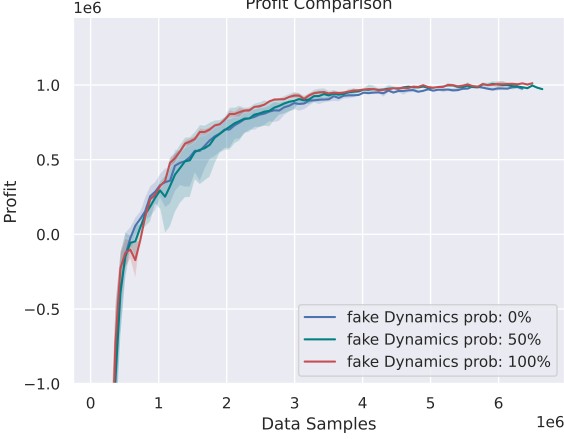

Figure 8: Training curves of CD-PPO with varied ratio of augmented data.

**Q2: Does dynamics augmentation improve the performance of the algorithm? If so, how much should we perturb the original data?**

We run similar experiments on environment N5-C100 with CD-PPO, in which the local simulator is ingested with a mixture of original dynamics data and LSTM generated data. The ratio of perturbed dynamics data varies from 0% to 100%. And we found that the algorithm turns out the best performance when we use fully generated data, as shown in Figure 8.

Table 5: The complete experiment results of all the variants.

| Env Scenario | N5-C50 | N5-C100 | N50-C500 | N50-C2000 | N100-C1000 | N100-C4000 |
|---|---|---|---|---|---|---|
| CD-PPO (ours) | 40.58±6.02 | **99.21±1.91** | 310.81±76.46 | **694.87±174.184** | **660.28±149.94** | **1297.75±124.52** |
| IPPO-IR(w/o context) | 40.37±4.89 | 92.41±2.78 | 235.09±60.61 | 689.27±48.92 | -2106.98±315.38 | -2223.11±2536.00 |
| MAPPO-IR(w/o context) | 39.32±15.53 | 94.70±18.84 | N/A | N/A | N/A | N/A |
| IPPO-IR | 43.33±3.30 | 91.38±3.57 | 250.03±58.38 | 545.86±459.71 | -1126.42±409.83 | 148.00±1017.47 |
| MAPPO-IR | 54.87±9.26 | 97.69±14.41 | N/A | N/A | N/A | N/A |
| IPPO(w/o context) | **74.11±1.55** | 97.89±6.65 | 164.43±143.01 | -1373.29±870.03 | -1768.19±1063.61 | -6501.42±6234.06 |
| MAPPO(w/o context) | 49.24±1.32 | 74.71±1.51 | N/A | N/A | N/A | N/A |
| IPPO | 63.22±13.75 | 92.90±13.36 | **366.74±89.58** | -1102.97±1115.69 | -669.83±1395.92 | -6019.28±9056.49 |
| MAPPO | 48.49±1.89 | 71.57±3.14 | N/A | N/A | N/A | N/A |
| Basestock(Static) | 17.4834 | 48.8944 | -430.0810 | -15.5912 | -173.39 | 410.59 |
| Basestock(Dynamic) | 33.9469 | 80.8602 | -408.1434 | 42.7092 | -22.05 | 493.32 |
| Basestock(Oracle) | 38.6207 | 97.7010 | -397.831 | 1023.6574 | 91.17 | 755.47 |

Table 6: Average number of samples needed by different algorithms to reach the median performance of the baselines on different environments.

| Env Scenario | N5-C50 | N5-C100 | N50-C500 | N50-C2000 | N100-C1000 | N100-C4000 |
|---|---|---|---|---|---|---|
| CD-PPO (ours) | ∞ | **522.80** | **5484.49** | **2996.87** | **47.14** | **60.07** |
| IPPO-IR-NC | ∞ | ∞ | ∞ | 3138.49 | ∞ | ∞ |
| MAPPO-IR-NC | ∞ | 711.97 | N/A | N/A | N/A | N/A |
| IPPO-IR | ∞ | ∞ | ∞ | 8483.04 | ∞ | 127.57 |
| MAPPO-IR | 708.10 | 987.27 | N/A | N/A | N/A | N/A |
| IPPO-NC | **588.63** | 806.56 | 9195.23 | ∞ | 539.26 | ∞ |
| MAPPO-NC | 1671.10 | ∞ | N/A | N/A | N/A | N/A |
| IPPO | 708.10 | 1298.40 | 12802.89 | ∞ | 191.88 | 1151.57 |
| MAPPO | 1671.10 | ∞ | N/A | N/A | N/A | N/A |

