# OpenReview forum: "Multi-Agent Reinforcement Learning with Shared Resources for Inventory Management"
_ICLR.cc/2023/Conference — Submitted to ICLR 2023_

### Official Review · Reviewer_Yg5s · 2022-10-21

**Confidence:** 3
**Correctness:** 2
**Technical Novelty And Significance:** 2
**Empirical Novelty And Significance:** 2
**Recommendation:** 5

**Clarity, Quality, Novelty And Reproducibility:**

Clarity:
Overall the paper is well-presented, except few places that I have mentioned in my detailed questions.

Quality:
Generally, the paper has the quality of an ICLR paper, it contributes to a sub-field of operations research, and studies an important problem

Novelty:
The proposed idea is not that novel, but fine enough if it can be backed up by numerical experiments.

Reproducibility:
The code of the paper is available at the review stage. Although I did not run it, I assume it can be run to get the same results that are presented in the paper. There are some ambiguity in the paper in terms of reproducibility which I asked in my detailed questions.

**Strength And Weaknesses:**

Strength:
- A new RL algorithm to consider the capacity constraint in single stage inventory optimization problem.
- The proposed algorithm works better than some existing classical and RL-based approaches in problems with up to 1000 items.

Weakness:
- Lack of appropriate benchmarks
- Weak presentation on some sections

Here are the details of my questions:

Q1- At equation (4), you add a constraint on the arrival inventory if that results in passing the inventory capacity. The question is where do you assume the extra inventory stays? Do not you need to consider the holding cost, or some type of penalty cost for that?
Q1-1 This issue becomes worst if you consider a multi-echelon inventory optimization problem, in which each stage receives inventory from its predecessors, and early arrival is usually penalized with high costs.

Q2- At the end of page 4, the last line, do you mean $c^{-i}_t$? It is now $c^{-i}$.

Q3- I am not sure if the assumption of using an independent simulator for each SKU is a good idea. Specifically, it is mentioned that the multi-SKU simulator includes "replenishing orders fulfillment scheduling and transportation management" which I am not sure if they are necessary for your purpose. If you do not have those details in the single SKU simulator, you do not need to have them in the multi-SKU simulator too. Definitely computation-wise it is helpful, but assume a store that chooses to purchase bulky items that can sell them very quickly, like the garden items for the early spring. Those items heavily affect the overall used capacity and the simulator cannot ignore them.


Q4- "MAPPO learns a centralized critic and therefore becomes intractable due to the large state-action space when the number of SKUs is greater or equal to 50". I would not use this baseline since it cannot scale as you mentioned. [1] uses an attention method to create the state input for the critic method to address this issue.

Q5- The details of the chosen "values" like the prices, the costs, and the leading time were not available in the actual dataset. Can you please add those in an appendix in the paper? Without those, it is not possible to replicate your result.

Q6- I did not understand why you need equation 8 and the LSTM model to learn the transition function for the next step for capacity. I assume you can easily obtain that by inventory updating rules. Why do you need a predictor there? Can you please clarify?

Q7- The IPPO algorithm uses only $s_t$. I am not sure why it cannot use $c_t$. In reality, it is a piece of available information for all agents. I am not sure why you have not added that as part of the state.

Q8- How do you monitor/penalize if a solution results in a violation of capacity constraint?

Q9- I think the mean-field approach could be a good benchmark for your work since it has been tested many times on many-agent problems. See [2,3] for more details.

Q10- The current algorithm cannot be directly applied to a multi-echelon inventory optimization problem. It is not clear how the excess inventory (compared to the capacity of successor/predecessors) can be handled. Also, the performance of that is not known. Given that, I would emphasize that in each piece of the paper that this algorithm is for a single-echelon inventory optimization problem.

Q11- How do you consider the service level in your model? In other words, how do you enforce the attainment of a given service level for each item?

Q12- A viable benchmark for your work would be [4]. Although it does not consider fixed-order cost, you should be able to set that as a benchmark for your model when the fixed-order cost is zero and check your optimality gap. This comparison could determine the true value of your algorithm.


Minor comments:

M1- provide the the pseudocode -> provide the pseudocode
M2- There are several cited arxiv papers which some (that I know) are published at some venue. I would suggest you go over all the papers and update them all.


[1] Iqbal, Shariq, and Fei Sha. "Actor-attention-critic for multi-agent reinforcement learning." International conference on machine learning. PMLR, 2019.

[2]] Yang Y, Luo R, Li M, "Mean field multi-agent reinforcement learning". PMLR, 2018: 5571-5580.

[3] Caines, Huang and Malhame, "Mean field games", 2017. DOI 10.1007/978-3-319-27335-8_7-1

**Summary Of The Paper:**

A multi-agent RL algorithm is proposed to solve the multi-SKU inventory optimization in a single-echelon problem. The goal is to obtain the order quantity for each SKU while the overall capacity constraint is held and the total profit is maximized. The profit is calculated by getting the sell amount, purchase cost, holding cost, and fixed order cost. To define the MDP, the state definition for each SKU at each time involves in-transit quantity, the sale price, its historical actions, and its customer demands.
To manage the capacity constraint that all the SKUs try to get as much as possible, they propose to use a context variable which is consisted of all the capacity usage of all SKUs at the current time and some of the previous steps.
To build the simulator, it is assumed that the effect of SKU on the overall capacity is negligible, so they mostly run the simulator for each agent locally.

The weights of the policy and critic networks are shared among all agents. The action is one of {0, 1/3, 2/3, 1, 4/3 , 5/3 , 2, 5/2 , 3, 4,5,6,7,9,12} times the average of daily sales over the last two weeks. The reward is the local cost of the agent, and the state definition is as mentioned earlier.


**Summary Of The Review:**

Overall it is a good paper and studies an important problem. Although, there are some gray areas in the paper and the claims so that I do not recommend acceptance with the current version. Once the author address my concerns/questions, I raise my vote.

---

### Official Review · Reviewer_626R · 2022-10-24

**Confidence:** 3
**Correctness:** 2
**Technical Novelty And Significance:** 2
**Empirical Novelty And Significance:** 2
**Recommendation:** 3

**Clarity, Quality, Novelty And Reproducibility:**

The paper is nicely written, but while it presents a lot of details regarding background on stochastic games and the inventory management problem, it falls short on technical details regarding the training procedure and proposed algorithm. While more information is presented in the appendix, it is still not clear enough what is the exact procedure and models used.

Additionally, in the SG definition, the rewards of the agents are usually defined individually and the general objective is not to maximise the sum of individual rewards, SGs accommodate for more general settings:

Busoniu, L., Babuska, R., & De Schutter, B. (2006, December). Multi-agent reinforcement learning: A survey. In 2006 9th International Conference on Control, Automation, Robotics and Vision (pp. 1-6). IEEE.


An important related work that I think is highly relevant here concerns the influence-augmented local simulators:

Suau, M., He, J., Spaan, M. T. J., & Oliehoek, F. A. (2022). Influence-Augmented Local Simulators: a Scalable Solution for Fast Deep RL in Large Networked Systems. In K. Chaudhuri, S. Jegelka, L. Song, C. Szepesvari, G. Niu, & S. Sabato (Eds.), Proceedings of the 39th International Conference on Machine Learning (Vol. 162, pp. 20604-20624). (Proceedings of Machine Learning Research; Vol. 162). PMLR. https://proceedings.mlr.press/v162/suau22a.html


Q1. My first question is on the introduction of the SRSG. How does this setting compare to Dec-POMDPs and can't you model the problem using this framework?

Q2. Do I understand correctly that the work claims CD-PPO is a decentralised MARL approach? As far as I understand, it uses a centralised actor and critic, trained using all the local experiences of each agent. Given the nature of the problem, once the shared dynamics are captured and modelled separately, then one could see this as a single agent setting, where you aim to train an effective policy, that is then shared by all SKUs. Is this the case for this method?

Q3. How does CD-PPO differ from IPPO that would use all the individual agent trajectories to train one single network, augmented by the context information?

Q4. Can you clarify how the context dynamics are captured and used in the local simulator?

Q5. What is the motivation for the two techniques presented at the end of the Implementation paragraph? What issues did you encounter and had to mitigate using this strategy?


**Strength And Weaknesses:**

Strengths:
- This work addresses inventory management, an important sub-problem of supply-chain management
- It tries to bridge the gap between MARL methods and real-world applications with a large number of agents, addressing issues such as sample efficiency and computational costs for large simulators
- It demonstrates a manner of learning effective policies, by modelling and predicting separately the shared dynamics and using these predictions in the training procedure of the agents

Weaknesses:
- Insufficient details are presented for the creation of the local simulator and the context dynamics modelling
- There are also some clarity issues regarding CD-PPO and how exactly it is different from IPPO or MAPPO, how many networks it uses (more details below)
- The clarity issues obfuscate also the ability to judge the experimental evaluation, and the significance of the results



**Summary Of The Paper:**

In the context of multi-agent reinforcement learning, this work addresses the problem of inventory management, where a large number of stock keeping units need to learn how to optimally make replenishment decisions, under a shared inventory capacity. The work contributes the formulation of the shared-resource stochastic game (SRSG) to capture this problem setting and proposes to decouple the agents for a more efficient training procedure, by separately capturing and modelling the shared dynamics (i.e., the state of the shared inventory capacity). This allows the use of a variant of PPO, denoted as Context-aware decentralised PPO (CD-PPO), to obtain a sample efficient MARL solution for this problem.


**Summary Of The Review:**

The work addresses an important issue regarding bridging the gap between real-world large multi-agent systems and MARL approaches. However, due to a lack of technical details, it is difficult to judge the novelty and significance of the presented contributions. I cannot currently recommend the acceptance of this work.

---

### Official Review · Reviewer_cFPA · 2022-10-24

**Confidence:** 4
**Correctness:** 3
**Technical Novelty And Significance:** 3
**Empirical Novelty And Significance:** 3
**Recommendation:** 6

**Clarity, Quality, Novelty And Reproducibility:**

Given the detail by which the algorithms were written, and the easy to follow annotation, the paper and the findings of the paper might be reproducible if the algorithms were interpreted correctly.

**Strength And Weaknesses:**

This paper has an impressive and detailed account for the different algorithms used as well as the hyperparameters to make them function. What was missing was wrapping up the paper within the concluding remarks so that the reader can at a glance summate the totality of the research paper.

**Summary Of The Paper:**

In this paper, multi-agent reinforcement learning was used over dynamic programming to optimise inventory management. In this study, they proposed using Shared-Resource Stochastic Game to capture the problem structure in the inventory management where different agents interact with each other through competing for shared resources - an interesting approach for this type of problem. In addition to this, they propose a novel algorithm called Context-aware Decentralized PPO that leverages the shared-resource structure to solve the inventory management problem efficiently. and finally they performed extensive experiments to demonstrate that their method can achieve the performance on par with state-of-the-art MARL algorithms. In their study, they lack the founding conclusion but based on the intext questions, what was set out was achieved within the scope of the study

**Summary Of The Review:**

I recommend that the paper be accepted, but that a concluding section is added to the paper. As it currently stands, there is a lot of concluding thoughts missing from the paper - however impressive the algorithms and hyperparameters were.

---

### Official Review · Reviewer_cRM3 · 2022-10-25

**Confidence:** 4
**Correctness:** 2
**Technical Novelty And Significance:** 2
**Empirical Novelty And Significance:** 1
**Recommendation:** 5

**Clarity, Quality, Novelty And Reproducibility:**

Overall the authors justified the formulation of the problem and conducted experiments to showcase the superiority of the proposed CD-PPO methodology but at several places clarity was not provided which has been showcased in the weakness section. Results and discussion section need to be explored in detail showcasing the superior performance of CD-PPO with the parameters used in the MARL and the proposed algorithms. Especially with respect to sample efficiency, there are not enough evidences /rationale supporting its superior performance.
The authors have provided enough information to have the reproducibility.

**Strength And Weaknesses:**


Strengths:
1. Interesting to address inventory management problem through a Shared-resource Stochastic Game (SRSG)
2. Introduction of context awareness to avoid non-stationary problem and acceleration of sampling process in estimating and optimizing objective function.
3. Experimentation set up and baselining methods used in the study

Weakness:
1. Although the authors claim that the CD-PPO algorithm has outperformed MARL but it was indicated that it is due to the avoidance of non-stationarity by introducing the context but there are other methods like IPPO-IR with context showed a poor performance. This needs to be explored in detail.
2. The problem space considered was too small to be accepted in a premier conference like ICLR.
3. Clear standing and providing relevant real-world applications with shared-resource structure will enhance the readability of this article.
4. Rationale for the selection of MARL algorithms needs to be explored in detail.
5. There are several areas where the usage of language needs to be improved considerably. Some of the example are as follows:
a. In our setting, the constraint on the shared resources (such as the inventory capacity) couples the otherwise independent control for each SKU.



**Summary Of The Paper:**

In this paper, the authors have considered the problem of managing inventory on the replenishments for a number of SKUs to handle the supply and demand. This paper proposes Context-aware Decentralized PPO (CD-PPO) by formulating the problem as a Shared-Resource Stochastic Game (SRSG). The authors have conducted rigorous experiments that CD-PPO learns quickly and achieve a good performance with MARL algorithms.

**Summary Of The Review:**

Overall a good research article but needs a lot of improvement to be considered for publication in ICLR 2023.

---

### Decision · Program_Chairs · 2023-01-20

**Decision:**

Reject

**Justification For Why Not Higher Score:**

lack of author response and low evaluation score

**Justification For Why Not Lower Score:**

N/A

**Metareview: Summary, Strengths And Weaknesses:**

This paper presents a mult-agent reinforcement learning approach to inventory management, an important problem in supply chain management. Due to low evaluation scores and lack of author response, the recommendation for this paper is reject. I hope the authors find the reviewer comments helpful for improving the paper for publication at a future venue.